# Bias/Variance is not the same as Approximation/Estimation

**Gavin Brown**                                                    *gavin.brown@manchester.ac.uk*
*Department of Computer Science, The University of Manchester*

**Riccardo Ali**                                                              *rma55@cam.ac.uk*
*Department of Computer Science & Technology, The University of Cambridge*

**Reviewed on OpenReview:** *https://openreview.net/forum?id=4TnFbv16hK*

## Abstract

We study the relation between two classical results: the bias-variance decomposition, and the approximation-estimation decomposition. Both are important conceptual tools in Machine Learning, helping us describe the nature of model fitting. It is commonly stated that they are "closely related", or "similar in spirit". However, sometimes it is said they are equivalent. In fact they are different, but have subtle connections cutting across learning theory, classical statistics, and information geometry, that (very surprisingly) have not been previously observed. We present several results for losses expressible as a Bregman divergence: a broad family with a known bias-variance decomposition. Discussion and future directions are presented for more general losses, including the 0/1 classification loss.

## 1 Introduction

Geman et al. (1992) introduced the bias-variance decomposition to the Machine Learning community, and Vapnik & Chervonenkis (1974) introduced the approximation-estimation decomposition, founding the field of statistical learning theory. Both decompositions help us understand model fitting: referring to model size, and some kind of trade-off. The terms are often used interchangeably. And yet, they are different things. The approximation-estimation decomposition refers to models drawn from some function class $\mathcal{F}$, and considers an *excess risk*—that is, the risk above that of the Bayes model—breaking it into two components:

$$\textbf{excess risk} \;=\; \textbf{approximation error} \;+\; \textbf{estimation error}. \tag{1}$$

We might choose to increase the size of our function class, perhaps by adding more parameters to our model. In this situation it is commonly understood that the approximation error will decrease, and the estimation error will increase (Von Luxburg & Schölkopf, 2011), beyond a certain point resulting in over-fitting of the model. In contrast to the abstract notion of a "function class", the *bias-variance* decomposition considers the risk of real, trained models, in expectation over possible training sets. Assuming there is a unique correct response for each given input $\boldsymbol{x}$ (i.e., no noise) it breaks the expected risk into two components:

$$\textbf{expected risk} \;=\; \textbf{bias} \;+\; \textbf{variance}. \tag{2}$$

As we increase model size: the bias tends to decrease, and the variance tends to increase, again determining the degree of over-fitting. Recently, it has become apparent that this trade-off is not always simple, e.g. with over-parameterised models; but, the decomposition still holds even if a simple trade-off does not. We note that this decomposition, as used in the Machine Learning literature, concerns inference of the response/target variable—and not of the parameters, as is more common in classical statistics.

It is easy, and common, to conflate these decompositions. With a literature review (see Appendix C) one can observe innocent (but imprecise) statements such as, they are "similar in spirit", but also the more extreme (and incorrect/misleading) "the trade-off between estimation error and approximation error is often called the bias/variance trade-off". In contrast, we identify the precise relationships. We present detailed analysis for Bregman divergences in section 3, and offer discussion for more general losses in section 4.

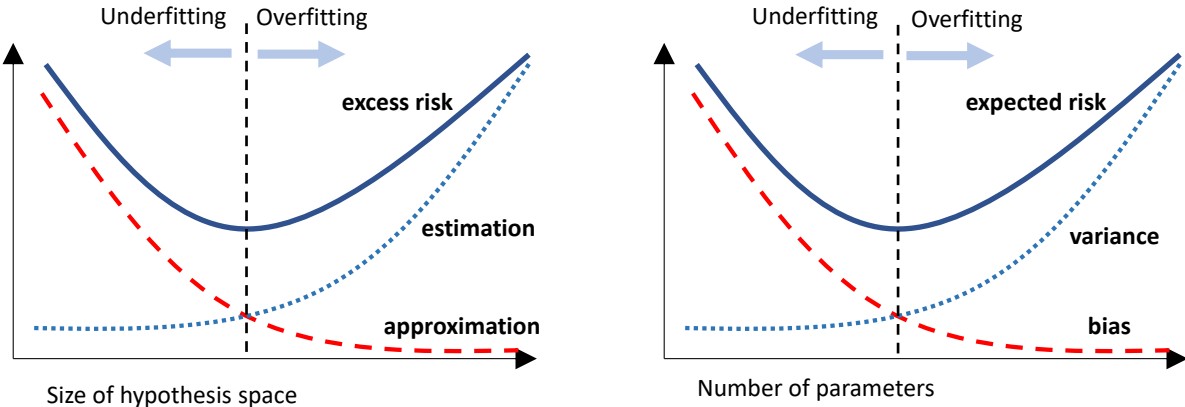

Figure 1: Two diagrams (same on left/right is intentional) illustrating how the approximation/estimation and bias/variance trade-offs are commonly described, and easily confused.

## 2 Background

We introduce notation and review the two decompositions. We introduce these ideas in an intentionally didactic/comprehensive manner, to avoid any possibility of confusion in terminology.

### 2.1 Preliminaries

Consider a standard supervised learning setup, where the task is to map from an input $\boldsymbol{x} \in \mathcal{X} \subseteq \mathbb{R}^d$ to an output $\boldsymbol{y} \in \mathcal{Y} \subseteq \mathbb{R}^k$, and assume there exists an unknown distribution $P(\boldsymbol{x}, \boldsymbol{y})$. This is achieved by learning a model $f$, which can also be seen as selecting a function $f$ from a restricted function class $\mathcal{F} \subset \mathcal{F}_{all}$, where $\mathcal{F}_{all}$ is the space of all measurable functions $\mathcal{X} \to \mathcal{Y}$. The discrepancy of $f(\boldsymbol{x})$ from the true $\boldsymbol{y}$ is quantified with a loss $\ell(\boldsymbol{y}, f(\boldsymbol{x}))$, which may or may not be symmetric. We define the *risk* of a model $f$ as,

$$R(f) := \mathbb{E}_{\boldsymbol{x}\boldsymbol{y}}[\ell(\boldsymbol{y}, f(\boldsymbol{x}))] = \int \ell(\boldsymbol{y}, f(\boldsymbol{x})) \; dP(\boldsymbol{x}, \boldsymbol{y}). \tag{3}$$

A *Bayes* model $\boldsymbol{y}^*$ is a (not necessarily unique) function which minimizes this quantity at each $\boldsymbol{x}$, i.e.

$$\boldsymbol{y}^* \in \arg\inf_{f \in \mathcal{F}_{all}} R(f), \tag{4}$$

where we follow Bach (2023, Prop 2.1) using the convention that the arginf returns a set if non-unique. We acknowledge a slight abuse of notation, using $\boldsymbol{y}^*$ as a function in $\mathcal{F}_{all}$ or a vector in $\mathbb{R}^k$ as needed—the intention will always be made clear from context. Given that we picked a restricted family $\mathcal{F} \subset \mathcal{F}_{all}$, we have no guarantee that it contains $\boldsymbol{y}^*$. A *best-in-family* model $f^*$ is defined similarly,

$$f^* \in \arg\inf_{f \in \mathcal{F}} R(f). \tag{5}$$

These are defined in terms of the true distribution $P(\boldsymbol{x}, \boldsymbol{y})$. In practice, we only have a finite sample: $n$ points $(\boldsymbol{x}_1, \boldsymbol{y}_1), ..., (\boldsymbol{x}_n, \boldsymbol{y}_n)$ each drawn i.i.d. from $P(\boldsymbol{x}, \boldsymbol{y})$, as a realisation of the random variable $D \sim P(\boldsymbol{x}, \boldsymbol{y})^n$. The *empirical risk* of a model $f \in \mathcal{F}$ is then defined:

$$R_{emp}(f) := \frac{1}{n} \sum_{i=1}^{n} \ell(\boldsymbol{y}_i, f(\boldsymbol{x}_i)), \tag{6}$$

and a model in $\mathcal{F}$ that minimizes this, known as an *empirical risk minimizer* (ERM), is defined:

$$\hat{f}_{erm} \in \arg\inf_{f \in \mathcal{F}} R_{emp}(f). \tag{7}$$

Note that $\hat{f}_{erm}$ is a random variable, as it is dependent on $D$. The empirical risks are the same for any two ERMs, but their population risks may be different. We can now cover the specifics for the two decompositions.

## 2.2 The Approximation-Estimation decomposition

The approximation-estimation decomposition is a seminal observation from the 1970s work of Vapnik and Chervonenkis, reviewed in Vapnik (1999). An excellent historical account can be found in Bottou (2013). The result deals with the *excess* risk $R(\hat{f}_{erm}) - R(\boldsymbol{y}^*)$, i.e. the risk of $\hat{f}_{erm}$ above that of the Bayes model, $\boldsymbol{y}^*$. The approximation-estimation decomposition, applicable for any loss $\ell$, breaks this into two terms:

$$\underbrace{R(\hat{f}_{erm}) - R(\boldsymbol{y}^*)}_{\textbf{excess risk}} \quad = \quad \underbrace{R(\hat{f}_{erm}) - R(f^*)}_{\textbf{estimation error}} \quad + \quad \underbrace{R(f^*) - R(\boldsymbol{y}^*)}_{\textbf{approximation error}}. \tag{8}$$

The approximation error is the additional risk due to using a restricted family $\mathcal{F}$, rather than the space of all functions $\mathcal{F}_{all}$. This is a systematic quantity, not dependent on any particular data sample. The estimation error is the additional risk due to our finite training data, when trying to find $f^* \in \mathcal{F}$. This a random variable, dependent on the particular data sample. There is a natural trade-off (see Figure 1, left) as we change the size of $\mathcal{F}$, keeping data size fixed. As we increase $|\mathcal{F}|$, approximation error will likely decrease (potentially to zero, if $\boldsymbol{y}^* \in \mathcal{F}$), but estimation error will increase, as it becomes harder to find $f^*$ in the larger space. The reason behind this is, in effect, the classical *multiple hypothesis testing* problem—we cannot reliably distinguish many hypotheses when our dataset is small. Bottou & Bousquet (2007) extended Equation 8, recognising that it is often intractable to find a global minimum of the training risk, and we can only have a sub-optimal model $\hat{f}$. An additional risk component then emerges, and the excess risk of $\hat{f}$ now decomposes into a sum of *optimisation* error, estimation error, and approximation error:

$$\underbrace{R(\hat{f}) - R(\boldsymbol{y}^*)}_{\textbf{excess risk of } \hat{f}} \quad = \quad \underbrace{R(\hat{f}) - R(\hat{f}_{erm})}_{\textbf{optimisation error}} \quad + \quad \underbrace{R(\hat{f}_{erm}) - R(f^*)}_{\textbf{estimation error}} \quad + \quad \underbrace{R(f^*) - R(\boldsymbol{y}^*)}_{\textbf{approximation error}}. \tag{9}$$

These three terms describe the learning process in abstract form: accounting respectively for the choice of learning algorithm, the quality/amount of data, and the capacity of the model family.

## 2.3 The Bias-Variance decomposition

A bias-variance decomposition involves the *expected* risk of a trained model $\hat{f}$, where the expectation $\mathbb{E}_D$ is over the random variable $D \sim P(\boldsymbol{x}, y)^n$, i.e., all possible training sets of a fixed size $n$. Focusing on a squared loss, and $y \in \mathbb{R}$, Geman et al. (1992) showed:

$$\underbrace{\mathbb{E}_D\left[\mathbb{E}_{\boldsymbol{x}y}[(y - \hat{f}(\boldsymbol{x}))^2]\right]}_{\textbf{expected risk}} = \underbrace{\mathbb{E}_{\boldsymbol{x}y}\left[(y - y^*)^2\right]}_{\textbf{noise}} + \underbrace{\mathbb{E}_{\boldsymbol{x}}\left[\left(y^* - \mathbb{E}_D[\hat{f}(\boldsymbol{x})]\right)^2\right]}_{\textbf{bias}} + \underbrace{\mathbb{E}_{\boldsymbol{x}}\left[\mathbb{E}_D[(\hat{f}(\boldsymbol{x}) - \mathbb{E}_D[\hat{f}(\boldsymbol{x})])^2]\right]}_{\textbf{variance}}. \tag{10}$$

where $y^* = \mathbb{E}_{y|\boldsymbol{x}}[y]$ is the Bayes-optimal prediction at each point $\boldsymbol{x}$. The bias is a systematic component, independent of any particular training sample, and commonly regarded as measuring the 'strength' of a model. The variance measures the sensitivity of $\hat{f}$ to changes in the training sample, independent of the true label $y$. The noise is a constant, independent of any model parameters. There is again a perceived trade-off with these terms (see Figure 1, right). As the size of the (un-regularised) model increases: bias *tends* to decrease, and variance *tends* to increase. However, the trade-off can be more complex (e.g. with over-parameterized models) and the exact dynamics are an open research issue.

**Bias-Variance decompositions hold for more than just squared loss.** In fact the same form holds for several other losses, including the broad family of *Bregman divergences* (Bregman, 1967).

**Definition 1 (Bregman divergence)** *For a convex set $\mathcal{Y} \subseteq \mathbb{R}^k$, let $\phi : \mathcal{Y} \to \mathbb{R}$ be a strictly convex function, continuously differentiable on the interior of $\mathcal{Y}$, which is assumed non-empty. The Bregman divergence $B_\phi : \mathcal{Y} \times \text{ri}(\mathcal{Y}) \to [0, \infty)$ is defined, for points $\boldsymbol{p} \in \mathcal{Y}$ and $\boldsymbol{q} \in \text{ri}(\mathcal{Y})$, as*

$$B_\phi(\boldsymbol{p}, \boldsymbol{q}) = \phi(\boldsymbol{p}) - \phi(\boldsymbol{q}) - \langle \nabla\phi(\boldsymbol{q}), \boldsymbol{p} - \boldsymbol{q} \rangle \tag{11}$$

The function $\phi$ is conventionally referred to as a 'generator' function—the choice of which leads to different well-known losses, e.g. $\phi(y) = y^2$ gives $B_\phi(y, f) = (y - f)^2$. We refer the reader to Banerjee et al. (2005b) for an excellent tutorial on Bregman divergences.

Key to understanding generalised bias-variance decompositions is the notion of a *centroid*. Nielsen & Nock (2009) provide a thorough characterisation for the centroids of Bregman divergences.

**Definition 2 (Left-sided Bregman centroid)** *Assume a Bregman divergence $B_\phi : \mathcal{Y} \times \mathrm{ri}(\mathcal{Y}) \to [0, \infty)$, with generator $\phi : \mathcal{Y} \to \mathbb{R}$. Define a set of models of the form $\hat{f} : \mathcal{X} \to \mathrm{ri}(\mathcal{Y})$, induced by a random variable $D$, then the left-sided Bregman centroid of the model distribution is:*

$$\mathring{f}_\phi(\boldsymbol{x}) := \underset{\boldsymbol{z} \in \mathcal{Y}}{\arg\min} \ \mathbb{E}_D \left[ B_\phi(\boldsymbol{z}, \hat{f}(\boldsymbol{x})) \right] = [\nabla \phi]^{-1} \left( \mathbb{E}_D \left[ \nabla \phi(\hat{f}(\boldsymbol{x})) \right] \right). \tag{12}$$

If we choose $\phi(y) = y^2$, then $B_\phi(y, f) = (y - f)^2$, and $\mathring{f}_\phi(\boldsymbol{x}) = \mathbb{E}_D[\hat{f}(\boldsymbol{x})]$, but this is not always the case. The left[1] centroid is in general a quasi-arithmetic mean, is unique (Nielsen & Nock, 2009, Theorem 3.2), and guaranteed to exist (Nielsen & Nock, 2020, Theorem 1). Examples of Bregman centroids are below.

Table 1: Examples of Bregman divergences, with corresponding left centroids.

| Name | Domain | $B_\phi(\boldsymbol{y}, \hat{f}(\boldsymbol{x}))$ | Centroid $\mathring{f}_\phi(\boldsymbol{x})$ |
|---|---|---|---|
| Squared | $y \in \mathbb{R}$ | $(y - \hat{f}(\boldsymbol{x}))^2$ | $\mathbb{E}_D[\hat{f}(\boldsymbol{x})]$ |
| KL-divergence | $\boldsymbol{y} \in \mathbb{R}^k, s.t. \sum_c y_c = 1$ | $D_{KL}(\boldsymbol{y} \ \|\| \ \hat{f}(\boldsymbol{x}))$ | $Z^{-1} \exp(\mathbb{E}_D[\ln \hat{f}(\boldsymbol{x})])$ |
| Ikatura-Saito | $y \in [0, \infty)$ | $\frac{y}{\hat{f}(\boldsymbol{x})} - \ln \frac{y}{\hat{f}(\boldsymbol{x})} - 1$ | $1/\mathbb{E}_D[\hat{f}(\boldsymbol{x})^{-1}]$ |

A bias-variance decomposition for Bregman divergences was shown by Pfau (2013), taking the form:

$$\underbrace{\mathbb{E}_D \left[ \mathbb{E}_{\boldsymbol{xy}}[B_\phi(\boldsymbol{y}, \hat{f}(\boldsymbol{x}))] \right]}_{\textbf{expected risk}} = \underbrace{\mathbb{E}_{\boldsymbol{xy}} \left[ B_\phi(\boldsymbol{y}, \boldsymbol{y}^*) \right]}_{\textbf{noise}} + \underbrace{\mathbb{E}_{\boldsymbol{x}} \left[ B_\phi(\boldsymbol{y}^*, \mathring{f}_\phi(\boldsymbol{x})) \right]}_{\textbf{bias}} + \underbrace{\mathbb{E}_{\boldsymbol{x}} \left[ \mathbb{E}_D[B_\phi(\mathring{f}_\phi(\boldsymbol{x}), \hat{f}(\boldsymbol{x}))] \right]}_{\textbf{variance}}. \tag{13}$$

where $\mathring{f}_\phi(\boldsymbol{x})$ is the left centroid, and $\boldsymbol{y}^* = \mathbb{E}_{\boldsymbol{y}|\boldsymbol{x}}[\boldsymbol{y}]$ is the Bayes model (Banerjee et al., 2005b, Prop. 1). We note that given $\hat{f} : \mathcal{X} \to \mathrm{ri}(\mathcal{Y})$, and the fact that $\nabla\phi$ is a homeomorphism (by the invariance of domain theorem), this implies $\mathring{f}_\phi(\boldsymbol{x}) \in \mathrm{ri}(\mathcal{Y})$. The bias/variance terms take *different functional forms* for each loss. This has a consequence for nomenclature: the term in Equation 10 is sometimes called "*squared bias*". But, the square is an artefact from using squared loss, not present in other cases, hence we use simply 'bias'. Note that the KL example implies a decomposition for the cross-entropy, since the two differ only by a constant. It is interesting to note that generalised decompositions only appeared in the ML community with Heskes (1998), but the idea seems to be known much earlier in statistics, e.g., Hastie & Tibshirani (1986, Eq. 19).

**Bias-Variance decompositions do not hold for all losses.** The approximation-estimation decomposition, Equation 9, applies for *any* loss. This is not the case for the bias-variance decomposition. For example, the form of Equation 13 does not hold for the 0/1 loss—in this case, the variance term becomes dependent on the label distribution (Friedman, 1997). Several authors proposed alternative decompositions (Wolpert, 1997; James & Hastie, 1997; Heskes, 1998; Domingos, 2000). It is interesting to note that the concept of the loss centroid also occurs in this literature, referred to as the 'systematic' or 'main' prediction (Geurts, 2002). The necessary and sufficient conditions for such a decomposition are an open research question.

## 2.4 Summary

These decompositions are *conceptual* tools to describe the nature of model fitting. They are by no means perfect reflections of the process, most especially in the context of over-parameterized models (Nagarajan & Kolter, 2019; Zhang et al., 2021). However, it is *extremely* common to see papers making the incorrect assumption/claim that the two are equivalent, or that one is a special case of the other. Our purpose with this work is to correct these false assumptions, identifying *precisely* how the two connect.

---

[1]Note that this is a called a *left* centroid because the minimization is over the first (left-hand) argument. The *right* centroid can be similarly defined by minimizing over the second argument, turning out to be simply $\mathbb{E}_D[\hat{f}(\boldsymbol{x})]$ for any valid $\phi$ (Banerjee et al., 2005a), which explains why the Bayes-optimal prediction is $\boldsymbol{y}^* = \mathbb{E}_{\boldsymbol{y}|\boldsymbol{x}}[\boldsymbol{y}]$ for any Bregman divergence.

## 3 Bias/Variance is not the same as Approximation/Estimation

By now it should be evident that these decompositions are related, but are not quite the same thing. Perhaps the most obvious difference is that they are on different quantities—the excess risk of an ERM, versus the expected risk of an arbitrary trained model. We now build a bridge between the two, using Bregman divergences to include a wide range of losses. We first define a concept, building on Definition 2, that we will refer to repeatedly in the coming sections: the 'centroid model'.

**Definition 3 (Centroid model)** *For a model $\hat{f}$ dependent on a random variable $D$, the centroid model $\mathring{f}_\phi$ is the aggregate model formed by taking the left Bregman centroid prediction at each possible $\boldsymbol{x}$. Note that whilst by definition $\mathring{f}_\phi \in \mathcal{F}_{all}$, there is no guarantee that $\mathring{f}_\phi \in \mathcal{F}$.*

We now observe that the estimation error involves $R(\hat{f}_{erm})$, making it a random variable dependent on $D$. We take the expectation with respect to $D$ and separate it into two, using the risk of the centroid model.

**Definition 4 (Estimation Bias, and Estimation Variance)** *For a Bregman divergence $B_\phi$, the expected estimation error can be decomposed to expose two terms: the estimation bias, and estimation variance.*

$$\underbrace{\mathbb{E}_D\left[R(\hat{f}_{erm}) - R(f^*)\right]}_{\textbf{expected estimation error}} = \underbrace{\mathbb{E}_D\left[R(\hat{f}_{erm}) - R(\mathring{f}_\phi)\right]}_{\textbf{estimation variance}} + \underbrace{R(\mathring{f}_\phi) - R(f^*)}_{\textbf{estimation bias}}. \tag{14}$$

The estimation variance measures the *random* variations of $\hat{f}_{erm}$ around the centroid model. The estimation bias measures the *systematic* difference between the centroid model and the best-in-family model. Using these concepts, we can present the relation between the two decompositions.

**Theorem 1 (Bias-Variance in terms of Approximation-Estimation)** *Given a Bregman divergence $B_\phi(\boldsymbol{y}, f(\boldsymbol{x}))$, the following decomposition of the bias and variance applies.*

$$\underbrace{\mathbb{E}_{\boldsymbol{x}}\left[B_\phi(\boldsymbol{y}^*, \mathring{f}_\phi(\boldsymbol{x}))\right]}_{\textbf{bias}} = \underbrace{R(f^*) - R(\boldsymbol{y}^*)}_{\textbf{approximation error}} + \underbrace{R(\mathring{f}_\phi) - R(f^*)}_{\textbf{estimation bias}} \tag{15}$$

$$\underbrace{\mathbb{E}_{\boldsymbol{x}}\left[\mathbb{E}_D[B_\phi(\mathring{f}_\phi(\boldsymbol{x}), \hat{f}(\boldsymbol{x}))]\right]}_{\textbf{variance}} = \underbrace{\mathbb{E}_D\left[R(\hat{f}) - R(\hat{f}_{erm})\right]}_{\textbf{optimisation error}} + \underbrace{\mathbb{E}_D\left[R(\hat{f}_{erm}) - R(\mathring{f}_\phi)\right]}_{\textbf{estimation variance}} \tag{16}$$

This confirms the premise of our work. Bias is *not* approximation error, and variance is *not* estimation error. It is not even the case that one is a special case of the other, as is sometimes stated. The true relation is more subtle. The approximation error is in fact just *one component of the bias*, and, the estimation error *contributes to both bias and variance.* The theorem above is illustrated in Figure 2.

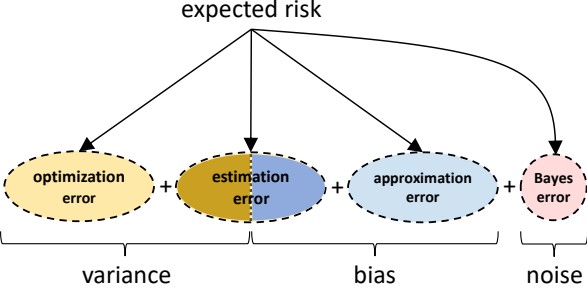

Figure 2: Illustration of Theorem 1. The bias is only partly determined by approximation error (i.e. choice of **model**), while the rest is due to expected estimation error (i.e. choice of **data**). Similarly, variation in data accounts for only part of the variance, and the rest is due to optimisation error (i.e. choice of **algorithm**).

## 4   Discussion

A simplistic description of bias and variance would say they are the error *'due to the model'* (bias) and the error *'due to the data'* (variance). Theorem 1 shows there is more nuance to understand. We now discuss the subtleties and implications of these results. For ease of referral we denote: $\mathcal{E}_{app}$ (approximation error), $\mathcal{E}_{est}$ (expected estimation error), $\mathcal{E}_{est(b)}$ (estimation bias), and $\mathcal{E}_{est(v)}$ (estimation variance).

### 4.1   The case of linear least squares

Assume a linear model $\hat{f}(\boldsymbol{x}) = \boldsymbol{x}^T \hat{\beta}$, and a squared loss. The parameters $\hat{\beta}$ are fit using a dataset $\{\mathbf{X}, \boldsymbol{y}\}$, where $\mathbf{X}$ is an $n \times d$ matrix, and $\boldsymbol{y}$ is a column vector of length $n$. As before, this training data is a realisation of the random variable $D$. The ridge regression solution is $\hat{\beta}_\lambda := [\mathbf{X}^T\mathbf{X} + \lambda\mathbf{I}]^{-1}\mathbf{X}^T\boldsymbol{y}$ where $\lambda = 0$ is OLS. The best-in-family model $f^*$, with $\mathcal{F} = \{f(\boldsymbol{x}) = \boldsymbol{x}^T\beta \mid \beta \in \mathbb{R}^d\}$, uses parameters $\beta_*$ which minimise the squared risk, i.e., $\beta_* := \arg\min_{\hat{\beta}} \mathbb{E}_{\boldsymbol{x}y}[(y - \boldsymbol{x}^T\hat{\beta})^2] = \mathbb{E}_{\boldsymbol{x}}[\boldsymbol{x}\boldsymbol{x}^T]^{-1}\mathbb{E}_{\boldsymbol{x}y}[\boldsymbol{x}y]$. Furthermore, due to the unbiasedness of OLS, $\mathbb{E}_D[\hat{\beta}_0] = \beta_*$. In this scenario, the bias-variance decomposition is,

$$\underbrace{\mathbb{E}_D\Big[\mathbb{E}_{\boldsymbol{x}y}\big[(y - \boldsymbol{x}^T\hat{\beta}_\lambda)^2\big]\Big]}_{expected\ risk} = \underbrace{\mathbb{E}_{\boldsymbol{x}y}\big[(y - y^*)^2\big]}_{noise} + \underbrace{\mathbb{E}_{\boldsymbol{x}}\big[(y^* - \mathbb{E}_D[\boldsymbol{x}^T\hat{\beta}_\lambda])^2\big]}_{bias} + \underbrace{\mathbb{E}_{\boldsymbol{x}}\Big[\mathbb{E}_D\big[(\boldsymbol{x}^T\hat{\beta}_\lambda - \mathbb{E}_D[\boldsymbol{x}^T\hat{\beta}_\lambda])^2\big]\Big]}_{variance}.$$
(17)

Hastie et al. (2017, Eq 7.14) describe[2] how the bias term decomposes more finely:

$$\underbrace{\mathbb{E}_{\boldsymbol{x}}\Big[(y^* - \mathbb{E}_D[\boldsymbol{x}^T\hat{\beta}_\lambda])^2\Big]}_{bias} = \underbrace{\mathbb{E}_{\boldsymbol{x}}\Big[(y^* - \boldsymbol{x}^T\beta_*)^2\Big]}_{\text{Hastie's 'model bias'}} + \underbrace{\mathbb{E}_{\boldsymbol{x}}\Big[(\boldsymbol{x}^T\beta_* - \mathbb{E}_D[\boldsymbol{x}^T\hat{\beta}_\lambda])^2\Big]}_{\text{Hastie's 'estimation bias'}}.$$
(18)

Though they refer to the first term as 'model bias', it turns out that is exactly equal to the approximation error for a linear model. Similarly, their 'estimation bias', whilst written differently, is exactly the estimation bias we have defined, but for a linear model. These observations are formalized in the following theorem.

**Theorem 2** *The "model bias/estimation bias" decomposition (Equation 18) is a special case of our bias decomposition (Equation 15) for the specific case of a linear model with squared loss.*

We also note that, due to the unbiasedness of OLS, we have both $\mathcal{E}_{opt} = 0$ and $\mathcal{E}_{est(b)} = 0$. Thus, for this special case, bias is equal to the approximation error, and variance is equal to (what remains of) the estimation error. The decompositions are (numerically) equivalent in the OLS scenario, but in general they are very different. This is most clear in the general case behaviour of the estimation bias, discussed next.

### 4.2   The bias is a flawed proxy for model capacity.

It is common to assume the bias is an indication of how simple/complex a model is—expected to be lower if the model has higher 'capacity'. But what is model 'capacity'? If we define it as the ability to minimize population risk, then the *ultimate* measure of model capacity is the *approximation error*. We see in Equation 15 that the bias contains exactly this, but also the *estimation bias*, which gives it some surprising dynamics. As detailed above, Hastie et al's observations were restricted to the linear case for analytic tractability. The linear model assumption meant that they were unable to observe a critical fact—that in the general case, the estimation bias $\mathcal{E}_{est(b)} = R(\mathring{f}_\phi) - R(f^*)$, can take *negative values,* i.e.

$$\mathbf{bias} \quad = \quad \underbrace{\Big[\substack{approximation \\ error}\Big]}_{\mathbf{always \geq 0}} \quad + \quad \underbrace{\Big[\substack{estimation \\ bias}\Big]}_{\mathbf{can\ be\ negative}} \tag{19}$$

To understand how this can be, we must accept the somewhat non-intuitive idea that the centroid model can be outside the hypothesis class $\mathcal{F}$, and thus we can have $R(\mathring{f}_\phi) < R(f^*)$. This can be trivially illustrated, with a simple regression stump evaluated by squared loss, in Figure 3.

---

[2]It is likely that Hastie et al were not the first to observe this, but it is a commonly known reference for the statement.

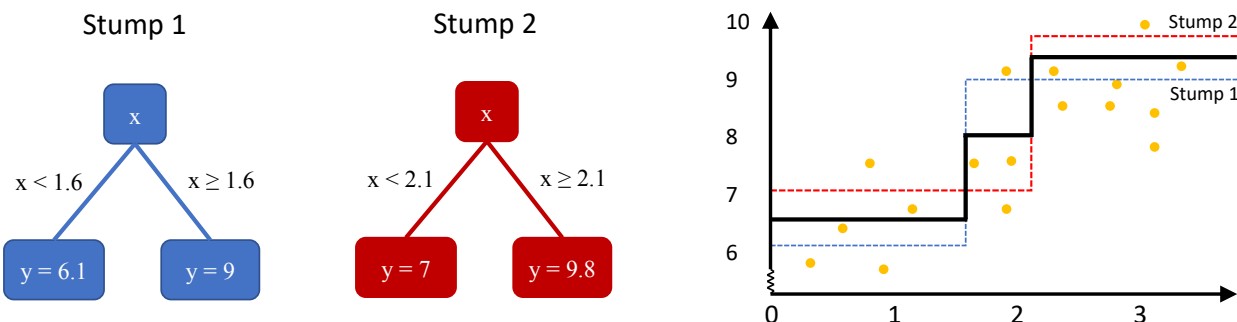

Figure 3: Two regression stumps (red/blue lines), and their centroid model (black line, arithmetic mean). Notice the centroid model is *outside* the hypothesis class, i.e. it cannot be represented as a single binary stump. As a result, the centroid model fits the data better than any $f \in \mathcal{F}$, and $\mathcal{E}_{est(b)}$ is negative.

The possibility of negative values here has significant implications. There are two ways in which bias can be zero. If $\mathcal{F}$ contains the Bayes model, then we might have $\mathcal{E}_{app} = \mathcal{E}_{est(b)} = 0$. But, there is another way. For some $\epsilon > 0$, we might have $\mathcal{E}_{app} = \epsilon$, and $\mathcal{E}_{est(b)} = -\epsilon$. In this case, the model family does *not* have sufficient capacity, since $\mathcal{E}_{app} > 0$. And yet, the bias is zero. Hence, the bias is a flawed proxy for the true model capacity. To illustrate this, we show experiments on a synthetic problem. Details in Appendix B.

Figure 4 shows results increasing the depth of a decision tree. The left panel shows excess risk, and the bias/variance components. We observe the classical bias/variance trade-off, including overfitting, as the depth increases beyond a certain point. It is notable that *the bias decreases to zero*, after depth 6. *Does this imply the model is 'unbiased', in the sense that it has sufficient capacity to capture the full data distribution?*

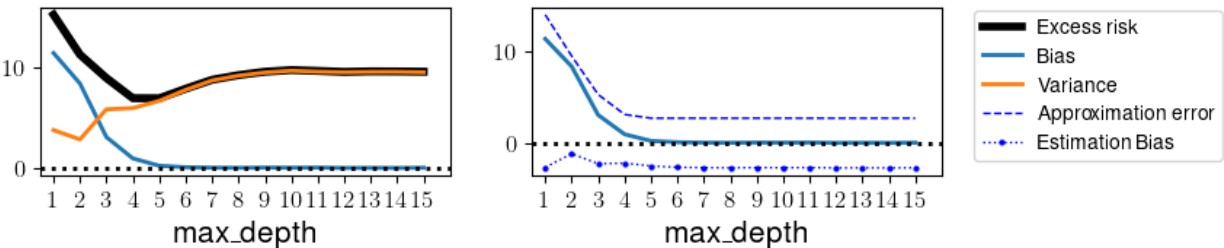

Figure 4: Risk components as we increase the depth of a regression tree.

The answer is no. A decomposition of the bias into two components (right panel) shows that the $\mathcal{E}_{app}$ is non-zero, i.e. the best possible model *cannot* achieve zero testing error. The cause of the bias going to zero is that $\mathcal{E}_{est(b)}$ is *negative*, hence the bias is not a good proxy for the true model capacity. Similar results are obtained with a $k$-nn regression (Figure 5), where increasing complexity corresponds to *decreasing k*.

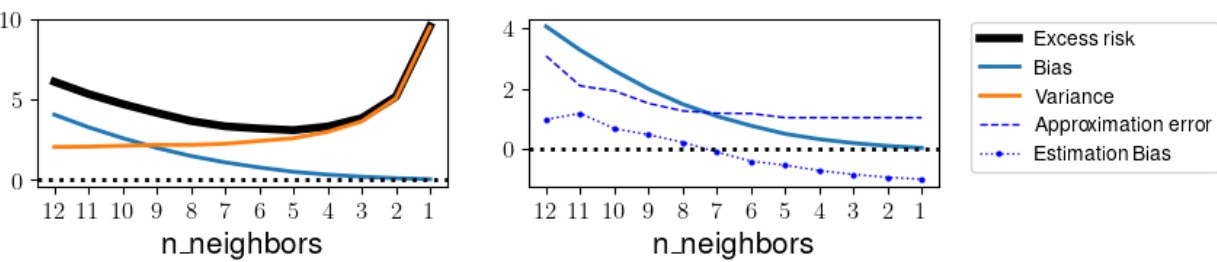

Figure 5: Risk components as we decrease the number of neighbours in a k-nn.

We can formally characterise this phenomenon, by studying the geometry of the hypothesis class $\mathcal{F}$. In particular, if the set $\mathcal{F}$ is *dual-convex* (Amari, 2008, Equation 32) with respect to $\phi$, then $\mathring{f}_\phi \in \mathcal{F}$, and hence estimation bias is guaranteed to be non-negative.

**Theorem 3 (Sufficient condition for a non-negative estimation bias.)** *If the hypothesis class $\mathcal{F}$ is dual-convex then the estimation bias is non-negative.*

A simple example of a non-dual convex set is the class of regression stumps evaluated by squared loss, where $\mathring{f}_\phi(\boldsymbol{x}) = \mathbb{E}_D[\hat{f}(\boldsymbol{x})]$, illustrated in Figure 3. A simple example of a dual-convex set is the class of Generalized Linear Models evaluated by their corresponding deviance measure.

**Theorem 4 (GLMs have non-negative estimation bias.)** *For a Bregman divergence with generator function $\phi$, define $\mathcal{F}$ as the set of all GLMs with inverse link $[\nabla\phi]^{-1}$ and natural parameters $\boldsymbol{\theta} \in \mathbb{R}^d$. Then, the estimation bias is non-negative.*

An example of this would be a logistic regression, $\hat{f}(\boldsymbol{x}) = [\nabla\phi]^{-1}(\hat{\boldsymbol{\theta}}^T\boldsymbol{x}) = 1/(1 + \exp(-\hat{\boldsymbol{\theta}}^T\boldsymbol{x}))$, which results from $\phi(f) = f \ln f + (1-f)\ln(1-f)$ and the binary KL is the corresponding Bregman divergence.

### 4.3 The estimation variance plays a role in double descent.

In recent literature, an increasing degree of over-parameterisation has been associated with a 'peaking' trend in the variance (Nakkiran, 2019; Yang et al., 2020), ultimately causing a *double descent* in the risk.

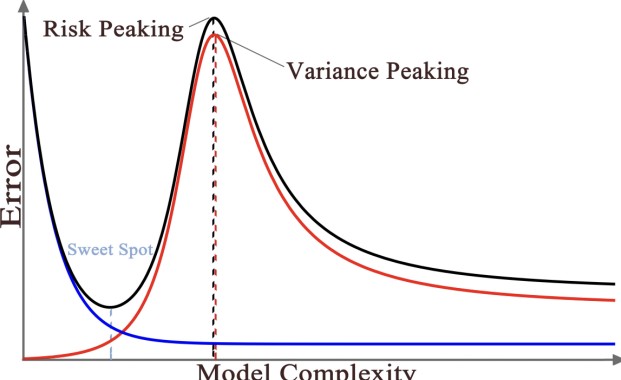

Figure 6: Illustration of double descent, caused by a 'peaking' variance (red line) and monotonically decreasing bias (blue line). Image credit Yang et al. (2020).

Such models often fit their training data perfectly (Belkin et al., 2019; Zhang et al., 2021), i.e., they *interpolate* the data. If we consider this in the context of Equation 16, we see that:

$$\mathbf{variance} \;=\; \begin{bmatrix} estimation \\ variance \end{bmatrix} \;+\; \underbrace{\begin{bmatrix} optimisation \\ error \end{bmatrix}}_{\approx\,0\ \mathbf{for\ interpolating\ models}}.$$

i.e., the optimization error is close to zero. This observed 'peaking' variance must therefore be primarily due to the *estimation variance*, $\mathcal{E}_{est(v)}$. Furthermore, very deep models are likely to be able to fit any function, i.e., their approximation error is zero. In these scenarios, the *only terms* remaining in the expected risk are $\mathcal{E}_{est(b)}$ and $\mathcal{E}_{est(v)}$. *Why* such models can push training error to zero, even on random labels, and still generalise well, remains an open question for modern machine learning (Zhang et al., 2021). Overall, we believe this warrants further study in the context of deep models.

### 4.4 New insights into the bias/variance trade-off.

In recent years, the relevance (and even existence) of a *trade-off* between bias and variance has been debated, with voices both against (Neal et al., 2018; Dar et al., 2021) and in favour (Witten, 2020). However with *estimation* bias and *estimation* variance, in certain circumstances, we observe that the trade-off is an indisputable *fact*. We first note that $\mathcal{E}_{est(b)} + \mathcal{E}_{est(v)} = \mathcal{E}_{est} \geq 0$. Thus, when one of these quantities is negative, the other is *forced* to be positive and lower bounded by the magnitude of the negative term. For example, when the estimation bias is negative (e.g. Figure 3), it obviously *reduces* the bias. However this implies that the variance increases, since $\mathcal{E}_{est(v)} \geq -\mathcal{E}_{est(b)}$, to satisfy the constraint. This is not the case for Hastie et al's terms, using a linear model / squared loss, since this is a (generalized) linear model and its corresponding deviance measure, as in Theorem 4. However, negative estimation bias is entirely possible in other scenarios, implying the variance (and indeed overall expected risk) is positive. This is an *unavoidable* trade-off. Clearly, other components of the bias/variance may mask this behaviour, making it less obvious.

### 4.5 What if a bias-variance decomposition doesn't hold?

As mentioned earlier, the form of Equation 13 does not hold for all losses, e.g., 0/1 loss. Many authors have proposed alternative 0/1 decompositions, each of which makes compromises and has different properties. The decomposition proposed by (James, 2003) applies for several losses, and links neatly to our work. They limit their commentary to *symmetric* losses—however, their conclusions do apply more generally, as we will see. As in our work, they rely on the notion of a centroid[3] model. For an arbitrary symmetric loss $\ell$, they could have defined the centroid in two ways: minimizing over the left or right argument:

$$\mathring{f}_{\text{LEFT}}(\boldsymbol{x}) := \underset{\boldsymbol{z} \in \mathcal{Y}}{\arg\min} \ \mathbb{E}_D\left[\ell(\boldsymbol{z}, \hat{f}(\boldsymbol{x}))\right], \qquad \mathring{f}_{\text{RIGHT}}(\boldsymbol{x}) := \underset{\boldsymbol{z} \in \mathcal{Y}}{\arg\min} \ \mathbb{E}_D\left[\ell(\hat{f}(\boldsymbol{x}), \boldsymbol{z})\right]. \qquad (20)$$

They chose the right centroid, which unfortunately masks the generality of their conclusions. This can be seen by instead adopting the *left centroid*, as we did for Bregman divergences. We now describe their framework, but using $\mathring{f} := \mathring{f}_{\text{LEFT}}$. From this point onwards, we assume a loss $\ell$ such that the centroid always exists, but make no assumptions on its uniqueness.

If $\ell$ was the 0/1 loss, $\mathring{f}$ is the *mode* of the distribution. For the absolute loss, it is the *median* value. If $\ell$ was a Bregman divergence, it is the left Bregman centroid, $\mathring{f}_\phi$, defined earlier. Using this, we define two terms:

$$\textbf{bias-effect} := R(\mathring{f}) - R(\boldsymbol{y}^*), \qquad (21)$$

$$\textbf{variance-effect} := \mathbb{E}_D\left[R(\hat{f}) - R(\mathring{f})\right]. \qquad (22)$$

These quantify the *effect on the risk* of using one predictor versus another. The *bias-effect* is the change in risk for the centroid model versus the Bayes model. The *variance-effect* is the change in risk for a model $\hat{f}$ versus the centroid model, averaged over the distribution of $D$. We then have the decomposition:

$$\underbrace{\mathbb{E}_D\left[R(\hat{f})\right]}_{\textbf{expected risk}} = \underbrace{R(\boldsymbol{y}^*)}_{\textbf{noise}} + \underbrace{R(\mathring{f}) - R(\boldsymbol{y}^*)}_{\textbf{bias-effect}} + \underbrace{\mathbb{E}_D[R(\hat{f}) - R(\mathring{f})]}_{\textbf{variance-effect}}, \qquad (23)$$

which can be verified by allowing terms on the right to cancel. James (2003) notes that with squared loss, the bias-effect is equal to the bias, and the variance-effect is equal to the variance: thus Equation 23 reduces to Geman et al. (1992). They state this relation holds for any *symmetric loss*, but in fact (as a trivial[4] corollary to Theorem 1) we see it holds for any Bregman divergence—which are all *asymmetric*, except squared loss. Thus, if $\ell$ is a Bregman divergence, Equation 23 reduces to Equation 13. We can relate the terms above to the approximation-estimation decomposition, using the same overall strategy as before.

---

[3]They refer to this as the 'systematic' part of the predictor, however we retain our terminology/notation for consistency.

[4]For bias, allow $R(f^*)$ to cancel on the right of Equation 15, and for variance, allow $R(\hat{f}_{erm})$ to cancel in Equation 16.

**Proposition 1 (Bias/Variance Effects, in terms of Approximation-Estimation)** *For any loss $\ell$, assuming a centroid model exists, we have the following decomposition of the bias-effect and variance-effect.*

$$\underbrace{R(\mathring{f}) - R(\boldsymbol{y}^*)}_{\text{bias-effect}} = \underbrace{R(f^*) - R(\boldsymbol{y}^*)}_{\text{approximation error}} + \underbrace{R(\mathring{f}) - R(f^*)}_{\text{estimation bias}}, \tag{24}$$

$$\underbrace{\mathbb{E}_D[R(\hat{f}) - R(\mathring{f})]}_{\text{variance-effect}} = \underbrace{\mathbb{E}_D\left[R(\hat{f}) - R(\hat{f}_{erm})\right]}_{\text{optimisation error}} + \underbrace{\mathbb{E}_D\left[R(\hat{f}_{erm}) - R(\mathring{f})\right]}_{\text{estimation variance}}. \tag{25}$$

And the full relation to our earlier observations can be illustrated as follows.

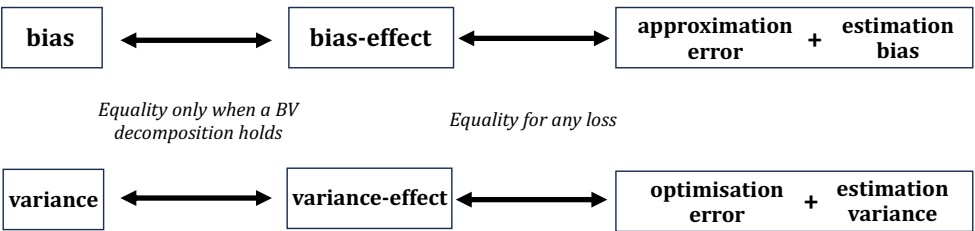

Figure 7: Relations between several decompositions that we have considered in this work.

As mentioned, for 0/1 loss the centroid is the *modal* value of the predictions. Taking the mode is effectively a weighted majority vote across the distribution of predictions from $\hat{f}$. Weighted voting classifiers have been extensively studied in the context of Boosting (Schapire, 2003), where it is well-known that a voted combination of weak (half-plane linear) models results a *non-linear* decision boundary. This implies $\mathring{f} \notin \mathcal{F}$, and thus again it is possible for estimation bias to be negative. Further characterisation of the terms in Figure 7, for the 0/1 loss, or indeed the general case of any loss, would therefore be desirable.

## 5 Conclusions

We analysed the precise connections between two seminal results that are often conflated: the bias-variance decomposition, and the approximation-estimation decomposition. Perhaps the most surprising aspect of this work was that it had not been explored before—two such foundational ideas, not previously connected. There are of course several excellent sources which do not conflate them, e.g., Györfi et al. (2002), but to the best of our knowledge there is no work comparing/contrasting the decompositions. In a literature review (see Appendix C), we found numerous sources stating the two were equivalent. This is false. The true relation, given by Theorem 1, is more intricate, and yielded interesting novel observations, including links to the phenomenon of double descent in deep learning. We focused on Bregman divergences, but also briefly considered the case of more general losses, where a bias-variance decomposition does not hold, e.g., 0/1 loss. In this case the geometry of such losses is not well-understood, leaving several open issues. In all cases, the *centroid model* turned out to be a key mathematical object in bridging the decompositions. We conjecture that further study of this object, and its role in generalisation, may yield yet deeper and interesting insights.

## Acknowledgements

Funding in direct support of this work: EPSRC EP/N035127/1 (LAMBDA project).

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

# Appendix

## A Proofs of Theorems

### A.1 Proof of Theorem 1 (Bias-Variance in terms of Approximation-Estimation).

We wish to prove the following statements:

$$\underbrace{\mathbb{E}_{\boldsymbol{x}}\left[B_\phi(\boldsymbol{y}^*, \mathring{f}_\phi(\boldsymbol{x}))\right]}_{\textbf{bias}} = \underbrace{R(f^*) - R(\boldsymbol{y}^*)}_{\textbf{approximation error}} + \underbrace{R(\mathring{f}_\phi) - R(f^*)}_{\textbf{estimation bias}} \tag{26}$$

$$\underbrace{\mathbb{E}_{\boldsymbol{x}}\left[\mathbb{E}_D\left[B_\phi(\mathring{f}_\phi(\boldsymbol{x}), \hat{f}(\boldsymbol{x}))\right]\right]}_{\textbf{variance}} = \underbrace{\mathbb{E}_D\left[R(\hat{f}) - R(\hat{f}_{erm})\right]}_{\textbf{optimisation error}} + \underbrace{\mathbb{E}_D\left[R(\hat{f}_{erm}) - R(\mathring{f}_\phi)\right]}_{\textbf{estimation variance}} \tag{27}$$

To show Equation 26, we note that the $R(f^*)$ terms cancel, so we just need to prove:

$$\mathbb{E}_{\boldsymbol{x}}\left[B_\phi(\boldsymbol{y}^*, \mathring{f}_\phi(\boldsymbol{x}))\right] = R(\mathring{f}_\phi) - R(\boldsymbol{y}^*). \tag{28}$$

The proof below builds on the *Bregman 3-point property* (Nielsen & Nock, 2009).

**Definition (Bregman three-point identity)** *The Bregman three-point property states, for any $p, q, r$,*

$$B_\phi(p, r) = B_\phi(p, q) + B_\phi(q, r) + \langle\ p - q\ ,\ \nabla\phi(q) - \nabla\phi(r)\ \rangle \tag{29}$$

We then have the following, where we apply the three-point property to $\boldsymbol{y}, \mathring{f}_\phi$, with $\boldsymbol{y}^*$ as the mid-point.

$$B_\phi(\boldsymbol{y}, \mathring{f}_\phi) = B_\phi(\boldsymbol{y}, \boldsymbol{y}^*) + B_\phi(\boldsymbol{y}^*, \mathring{f}_\phi) + \langle\ \boldsymbol{y} - \boldsymbol{y}^*\ ,\ \nabla\phi(\boldsymbol{y}^*) - \nabla\phi(\mathring{f}_\phi)\ \rangle \tag{30}$$

Take the expected value w/r $p(y|\boldsymbol{x})$ and the inner product term vanishes, since $\boldsymbol{y}^* = \mathbb{E}_{\boldsymbol{y}|\boldsymbol{x}}[y]$. Rearranging terms and further taking expectation w/r $\boldsymbol{x}$, we recover:

$$R(\mathring{f}_\phi) - R(\boldsymbol{y}^*) = \mathbb{E}_{\boldsymbol{x}}\left[B_\phi(\boldsymbol{y}^*, \mathring{f}_\phi)\right] \tag{31}$$

which is the desired result, proving Equation 26.

To show Equation 27, we follow a similar pattern. Take the 3-point property for $\boldsymbol{y}, \hat{f}$ with $\mathring{f}_\phi$ as the mid-point.

$$B_\phi(\boldsymbol{y}, \hat{f}) = B_\phi(\boldsymbol{y}, \mathring{f}_\phi) + B_\phi(\mathring{f}_\phi, \hat{f}) + \langle\ \boldsymbol{y} - \mathring{f}_\phi\ ,\ \nabla\phi(\mathring{f}_\phi) - \nabla\phi(\hat{f})\ \rangle \tag{32}$$

Take the expected value w/r $D$ and the inner product term vanishes, since $\nabla\phi(\mathring{f}_\phi) = \mathbb{E}_D\left[\nabla\phi(\hat{f})\right]$.

Rearranging terms and further taking expectation over $p(\boldsymbol{x})$, we recover:

$$\mathbb{E}_D\left[R(\hat{f}) - R(\mathring{f}_\phi)\right] = \mathbb{E}_{\boldsymbol{x}}\left[\mathbb{E}_D\left[B_\phi(\mathring{f}_\phi, \hat{f})\right]\right] \tag{33}$$

which is the desired result, completing the theorem.
∎

**Special case of Theorem 1 for squared loss.** The following presents the special case of squared loss, included for didactic purposes due to its ubiquity and links to the results for linear models. We wish to prove the following statements:

$$\mathbb{E}_{\boldsymbol{x}}\left[(\mathbb{E}_D[\hat{f}(\boldsymbol{x})] - \mathbb{E}_{y|\boldsymbol{x}}[y])^2\right] = \mathcal{E}_{app} + \mathcal{E}_{est(b)} \tag{34}$$

$$\mathbb{E}_{\boldsymbol{x}}\left[\mathbb{E}_D[(\hat{f}(\boldsymbol{x}) - \mathbb{E}_D[\hat{f}(\boldsymbol{x})])^2]\right] = \mathcal{E}_{opt} + \mathcal{E}_{est(v)} \tag{35}$$

$$\mathbb{E}_{\boldsymbol{x}y}[(y - \mathbb{E}_{y|\boldsymbol{x}}[y])^2] = R(y^*) \tag{36}$$

To show Equation 36, we simply note that $y^* = \mathbb{E}_{y|\boldsymbol{x}}[y]$, so the expression is true by definition.

To show Equation 34 we note, as an intermediate step, that:

$$\mathcal{E}_{app} + \mathcal{E}_{est(b)} = \Big(R(f^*) - R(y^*)\Big) + \Big(R(\mathbb{E}_D[\hat{f}]) - R(f^*)\Big) = R(\mathbb{E}_D[\hat{f}]) - R(y^*). \tag{37}$$

We then have the following, again using the definition of $y^*$.

$$
\begin{aligned}
R(\mathbb{E}_D[\hat{f}]) - R(y^*) &= \mathbb{E}_{\boldsymbol{x}y}\left[(\mathbb{E}_D[\hat{f}] - y)^2\right] - \mathbb{E}_{\boldsymbol{x}y}\left[(y - \mathbb{E}_{y|\boldsymbol{x}}[y])^2\right]. \\
&= \mathbb{E}_{\boldsymbol{x}y}\left[\left(\mathbb{E}_D[\hat{f}]\right)^2 - 2y\mathbb{E}_D[\hat{f}] - \mathbb{E}_{y|\boldsymbol{x}}[y]^2 + 2y\mathbb{E}_{y|\boldsymbol{x}}[y]\right] \\
&= \mathbb{E}_{\boldsymbol{x}}\left[\left(\mathbb{E}_D[\hat{f}]\right)^2 - 2\mathbb{E}_{y|\boldsymbol{x}}[y]\mathbb{E}_D[\hat{f}(\boldsymbol{x})] - \mathbb{E}_{y|\boldsymbol{x}}[y]^2 + 2\mathbb{E}_{y|\boldsymbol{x}}[y]^2\right] \\
&= \mathbb{E}_{\boldsymbol{x}}\left[\left(\mathbb{E}_D[\hat{f}]\right)^2 - 2\mathbb{E}_{y|\boldsymbol{x}}[y]\mathbb{E}_D[\hat{f}(\boldsymbol{x})] + \mathbb{E}_{y|\boldsymbol{x}}[y]^2\right] \\
&= \mathbb{E}_{\boldsymbol{x}}\left[(\mathbb{E}_D[\hat{f}] - \mathbb{E}_{y|\boldsymbol{x}}[y])^2\right],
\end{aligned}
$$

which is the bias, and the desired result.

To show Equation 35, we follow a similar pattern. From definitions:

$$\mathcal{E}_{opt} + \mathcal{E}_{est(v)} = \mathbb{E}_D\left[R(\hat{f}) - R(\hat{f}_{erm})\right] + \mathbb{E}_D\left[R(\hat{f}_{erm}) - R(\mathbb{E}_D[\hat{f}])\right] = \mathbb{E}_D\left[R(\hat{f}) - R(\mathbb{E}_D[\hat{f}])\right]. \tag{38}$$

We then have the following.

$$
\begin{aligned}
\mathbb{E}_D\left[R(\hat{f}) - R(\mathbb{E}_D[\hat{f}])\right] &= \mathbb{E}_D\left[\mathbb{E}_{\boldsymbol{x}y}\left[(\hat{f} - y)^2\right] - \mathbb{E}_{\boldsymbol{x}y}\left[(\mathbb{E}_D[\hat{f}] - y)^2\right]\right] \\
&= \mathbb{E}_D\left[\mathbb{E}_{\boldsymbol{x}y}\left[\hat{f}^2 - 2y\hat{f} - \mathbb{E}_D[\hat{f}]^2 + 2y\mathbb{E}_D[\hat{f}]\right]\right] \\
&= \mathbb{E}_{\boldsymbol{x}y}\left[\mathbb{E}_D[\hat{f}^2] - 2y\mathbb{E}_D[\hat{f}] - \mathbb{E}_D[\hat{f}]^2 + 2y\mathbb{E}_D[\hat{f}]\right] \\
&= \mathbb{E}_{\boldsymbol{x}}\left[\mathbb{E}_D[\hat{f}^2] - \mathbb{E}_D[\hat{f}]^2\right] \\
&= \mathbb{E}_{\boldsymbol{x}}\left[\mathbb{E}_D[(\hat{f} - \mathbb{E}_D[\hat{f}])^2]\right]
\end{aligned}
$$

where the final step is the standard definition of variance, giving the desired result.

∎

**A.2   Proof of Theorem 2 (Hastie's model bias / estimation bias is a special case).**

Hastie et al. (2017, Equation 7.14) observe:

$$\underbrace{\mathbb{E}_{\boldsymbol{x}}\Big[(y^* - \mathbb{E}_D[\boldsymbol{x}^T\hat{\beta}_\lambda])^2\Big]}_{\text{bias}} = \underbrace{\mathbb{E}_{\boldsymbol{x}}\Big[(y^* - \boldsymbol{x}^T\beta_*)^2\Big]}_{\text{Hastie's 'model bias'}} + \underbrace{\mathbb{E}_{\boldsymbol{x}}\Big[(\boldsymbol{x}^T\beta_* - \mathbb{E}_D[\boldsymbol{x}^T\hat{\beta}_\lambda])^2\Big]}_{\text{Hastie's 'estimation bias'}}. \tag{39}$$

We will show this is a special case of our Equation 15, restated here for $y \in \mathbb{R}$:

$$\underbrace{\mathbb{E}_{\boldsymbol{x}}\Big[B_\phi(y^*, \mathring{f}_\phi(\boldsymbol{x}))\Big]}_{\text{bias}} = \underbrace{R(f^*) - R(y^*)}_{\text{approximation error}} + \underbrace{R(\mathring{f}_\phi) - R(f^*)}_{\text{estimation bias}} \tag{40}$$

We first show Hastie's "model bias" is the approximation error of a linear model using squared loss, i.e.,

$$\underbrace{R(f^*) - R(y^*)}_{\substack{\text{approximation error} \\ \text{(generic form)}}} = \underbrace{\mathbb{E}_{\boldsymbol{x}}\Big[(y^* - \boldsymbol{x}^T\beta_*)^2\Big]}_{\substack{\text{Hastie's 'model bias'} \\ \text{(squared loss, linear model)}}}. \tag{41}$$

For any Bregman divergence $B_\phi$, the approximation error $R(f^*) - R(y^*)$ can be written as follows.

$$R(f^*) - R(y^*) = \mathbb{E}_{\boldsymbol{x}y}[B_\phi(y^*, f^*(\boldsymbol{x})]. \tag{42}$$

i.e. the approximation error is equal to the Bregman divergence of $y^*$ from $f^*(\boldsymbol{x})$, in expectation over $P(\boldsymbol{x}y)$.

**Proof sketch.** Use the 3-point theorem in exactly the same manner as in the proof of Theorem 1, i.e. between $y, f^*$ with $y^*$ as the mid-point:

$$B_\phi(y, f^*) = B_\phi(y, y^*) + B_\phi(y^*, f^*) + \langle\, y - y^* \,,\, \nabla\phi(y^*) - \nabla\phi(f^*) \,\rangle. \tag{43}$$

Then take expectation successively over $P(y|\boldsymbol{x})$ then $P(\boldsymbol{x})$. The inner product term is zero, since $\mathbb{E}_{y|\boldsymbol{x}}[y] = y^*$. For a linear model, $f^*(\boldsymbol{x}) = \boldsymbol{x}^T\beta_*$. Rearrange the remaining terms: using squared loss, we have the result.

The result for estimation bias is proven similarly. For squared loss, $\mathring{f}_\phi = \mathbb{E}_D[\boldsymbol{x}^T\hat{\beta}_\lambda]$, so we will show:

$$\underbrace{R(\mathring{f}_\phi) - R(f^*)}_{\substack{\text{estimation bias} \\ \text{(generic form)}}} = \underbrace{\mathbb{E}_{\boldsymbol{x}}\Big[(\boldsymbol{x}^T\beta_* - \mathbb{E}_D[\boldsymbol{x}^T\hat{\beta}_\lambda])^2\Big]}_{\substack{\text{Hastie's estimation bias} \\ \text{(squared loss, linear model)}}}. \tag{44}$$

**Proof sketch.** Estimation bias can be written (using the Bregman 3-point theorem) as so:

$$B_\phi(y, \mathring{f}_\phi) = B_\phi(y, f^*) + B_\phi(f^*, \mathring{f}_\phi) + \langle\, y - f^* \,,\, \nabla\phi(f^*) - \nabla\phi(\mathring{f}_\phi) \,\rangle. \tag{45}$$

We take squared loss and a linear model, $\hat{f}(\boldsymbol{x}) = \boldsymbol{x}^T\hat{\beta}_\lambda$, then take expectation over $P(\boldsymbol{x}, y)$, and we have,

$$\mathbb{E}_{\boldsymbol{x}y}[(y - \mathbb{E}_D[\boldsymbol{x}^T\hat{\beta}_\lambda])^2] = \mathbb{E}_{\boldsymbol{x}y}[(y - \boldsymbol{x}^T\beta_*)^2] + \mathbb{E}_{\boldsymbol{x}y}[(\boldsymbol{x}^T\beta_* - \mathbb{E}_D[\boldsymbol{x}^T\hat{\beta}_\lambda])^2], \tag{46}$$

$$R(\mathring{f}_\phi) - R(f^*) = \mathbb{E}_{\boldsymbol{x}y}[(\boldsymbol{x}^T\beta_* - \mathbb{E}_D[\boldsymbol{x}^T\hat{\beta}_\lambda])^2]. \tag{47}$$

where we note that in Equation 45, the cross term $\mathbb{E}_{\boldsymbol{x}y}\Big[\big(y - \boldsymbol{x}^T\beta_*\big)\big(\nabla\phi(\boldsymbol{x}^T\beta_*) - \nabla\phi(\mathbb{E}_D[\boldsymbol{x}^T\hat{\beta}_\lambda])\big)\Big] = 0$.

This can be shown as follows. We note that $\nabla\phi(z) = 2z$, and $\beta_* = \mathbb{E}_{\boldsymbol{x}}[\boldsymbol{x}\boldsymbol{x}^T]^{-1}\mathbb{E}_{\boldsymbol{x}y}[\boldsymbol{x}y]$.

$$\mathbb{E}_{\boldsymbol{x}y}\Big[\big(y - \boldsymbol{x}^T\beta_*\big)\big(\nabla\phi(\boldsymbol{x}^T\beta_*) - \nabla\phi(\mathbb{E}_D[\boldsymbol{x}^T\hat{\beta}_\lambda])\big)\Big] \tag{48}$$

$$= \mathbb{E}_{\boldsymbol{x}y}\Big[(y - \boldsymbol{x}^T\beta_*) \cdot 2\boldsymbol{x}^T\beta_* - (y - \boldsymbol{x}^T\beta_*) \cdot 2\mathbb{E}_D[\boldsymbol{x}^T\hat{\beta}_\lambda]\Big] \tag{49}$$

$$= 2\underbrace{\mathbb{E}_{\boldsymbol{x}y}\Big[(y - \boldsymbol{x}^T\beta_*) \cdot \boldsymbol{x}^T\beta_*\Big]}_{=0} - 2\underbrace{\mathbb{E}_{\boldsymbol{x}y}\Big[(y - \boldsymbol{x}^T\beta_*) \cdot \mathbb{E}_D[\boldsymbol{x}^T\hat{\beta}_\lambda]\Big]}_{=0} \tag{50}$$

Taking the first term, ignoring the constant 2, we have:

$$\mathbb{E}_{\boldsymbol{x}y}\left[(y - \boldsymbol{x}^T\beta_*) \cdot \boldsymbol{x}^T\beta_*\right] = \mathbb{E}_{\boldsymbol{x}y}\left[y\boldsymbol{x}^T - \beta_*^T\boldsymbol{x}\boldsymbol{x}^T\right] \cdot \beta_* \tag{51}$$

$$= \left(\mathbb{E}_{\boldsymbol{x}y}[y\boldsymbol{x}^T] - \beta_*^T\mathbb{E}_{\boldsymbol{x}}[\boldsymbol{x}\boldsymbol{x}^T]\right) \cdot \beta_* \tag{52}$$

$$= \left(\mathbb{E}_{\boldsymbol{x}y}[y\boldsymbol{x}^T] - \left[\mathbb{E}_{\boldsymbol{x}}[\boldsymbol{x}\boldsymbol{x}^T]^{-1}\mathbb{E}_{\boldsymbol{x}y}[\boldsymbol{x}y]\right]^T\mathbb{E}_{\boldsymbol{x}}[\boldsymbol{x}\boldsymbol{x}^T]\right) \cdot \beta_* \tag{53}$$

$$= \left(\mathbb{E}_{\boldsymbol{x}y}[y\boldsymbol{x}^T] - \mathbb{E}_{\boldsymbol{x}y}[\boldsymbol{x}y]^T\mathbb{E}_{\boldsymbol{x}}[\boldsymbol{x}\boldsymbol{x}^T]^{-1}\mathbb{E}_{\boldsymbol{x}}[\boldsymbol{x}\boldsymbol{x}^T]\right) \cdot \beta_* \tag{54}$$

$$= \left(\mathbb{E}_{\boldsymbol{x}y}[y\boldsymbol{x}^T] - \mathbb{E}_{\boldsymbol{x}y}[y\boldsymbol{x}^T]\right) \cdot \beta_* \tag{55}$$

$$= 0. \tag{56}$$

Follow the same steps for the second term, with $\mathbb{E}_D[\hat{\beta}_\lambda]$ in place of $\beta_*$.

∎

### A.3 Proof of Theorem 3 (Sufficient condition for a non-negative estimation bias).

To prove Theorem 3, we demonstrate that under a certain condition, $\mathring{f}_\phi \in \mathcal{F}$, which implies $R(\mathring{f}_\phi) \geq R(f^*)$, and therefore $R(\mathring{f}_\phi) - R(f^*) \geq 0$. We use the following definition, due to Amari (2008, Equation 32).

**Definition 5 (Dual convex set)** *Let $\phi$ be a strictly convex function. A set $\mathcal{F}$ is dually convex with respect to $\phi$ iff, for any pair of points $f, g \in \mathcal{F}$ and for all $\lambda \in [0, 1]$*

$$\lambda\nabla\phi(f) + (1-\lambda)\nabla\phi(g) \in \mathcal{F}$$

i.e. the set $\mathcal{F}$ is dual-convex iff it is convex in its dual coordinate representation.

An arbitrary set $\mathcal{C}$ is convex iff for any random variable $X$ defined over elements of $\mathcal{C}$, its expectation is also in $\mathcal{C}$, i.e. $\mathbb{E}[X] \in \mathcal{C}$.

Therefore, for a dual convex set $\mathcal{F}$, we have that the point $\mathbb{E}_D[\nabla\phi(f)] \in \mathcal{F}$. The primal coordinate representation of this point, $\nabla\phi^{-1}(\mathbb{E}_D[\nabla\phi(f)])$, is also a member of $\mathcal{F}$, i.e. $\mathring{f}_\phi \in \mathcal{F}$, proving the theorem.
∎

### A.4 Proof of Theorem 4 (GLMs have non-negative estimation bias).

We demonstrate that $\mathcal{E}_{est(b)} \geq 0$ if $\hat{f}$ is a GLM of a particular form. We give two proofs: a direct one and one that makes use of Theorem 3.

**Direct proof.** The estimation bias is defined:

$$\mathcal{E}_{est(b)} = R(\mathring{f}_\phi) - R(f^*). \tag{57}$$

This involves the definition of the centroid prediction, which for a Bregman divergence is,

$$\mathring{f}_\phi(\boldsymbol{x}) := [\nabla\phi]^{-1}\left(\mathbb{E}_D\left[\nabla\phi(\hat{f}(\boldsymbol{x}))\right]\right). \tag{58}$$

Given a Bregman divergence with generator $\phi$, define $\mathcal{F}$ as the class of GLMs with inverse link $[\nabla\phi]^{-1}$, parameterised by $\boldsymbol{\theta} \in \mathbb{R}^d$. In this case, each $\hat{f} \in \mathcal{F}$ takes the form:

$$\hat{f}(\boldsymbol{x}) := [\nabla\phi]^{-1}\left(\boldsymbol{\theta}^T\boldsymbol{x}\right), \tag{59}$$

where $\boldsymbol{\theta}$ are the natural parameters. Substituting this into the centroid prediction gives us,

$$
\begin{aligned}
\mathring{f}_\phi(\boldsymbol{x}) &= [\nabla\phi]^{-1}\left(\mathbb{E}_D\left[\nabla\phi\left([\nabla\phi]^{-1}\left(\boldsymbol{\theta}^T\boldsymbol{x}\right)\right)\right]\right), \\
&= [\nabla\phi]^{-1}\left(\mathbb{E}_D\left[\boldsymbol{\theta}\right]^T\boldsymbol{x}\right).
\end{aligned}
\tag{60}
$$

Since $\mathbb{E}_D[\boldsymbol{\theta}]$ is within the convex hull of the distribution of $\boldsymbol{\theta}$ induced by $D$, the centroid prediction is the same form of GLM as $\hat{f}(\boldsymbol{x})$, for all $\boldsymbol{x}$, and therefore the centroid model $\mathring{f}_\phi \in \mathcal{F}$. Then, since by definition $f^*$ is the risk minimizer in $\mathcal{F}$, we must have that $R(\mathring{f}_\phi) \geq R(f^*)$, and therefore Equation 57 is non-negative. ∎

**Proof using Theorem 3.** To show that the estimation bias is non-negative, it suffices to show that the class of GLMs of a particular form is *dually-convex*. We verify that the property of dual-convexity holds. Define $\mathcal{F} = $ GLMs with inverse link $\nabla\phi^{-1}$.

By definition, if $f \in \mathcal{F}$, it is parameterised by a vector $\boldsymbol{\theta}$ as follows: $f(\boldsymbol{x}) = \nabla\phi^{-1}(\boldsymbol{\theta}^T\boldsymbol{x})$.

Let $h$ be the function, expressed in primal coordinates, corresponding to the convex combination of two arbitrary GLMs in their dual coordinates, i.e. $h = \nabla\phi^{-1}(\lambda\nabla\phi(f) + (1-\lambda)\nabla\phi(g))$, with $\lambda \in [0,1]$, and with $f$ and $g$ two GLMs $f = \nabla\phi^{-1}(\boldsymbol{\theta}^T\boldsymbol{x})$ and $g = \nabla\phi^{-1}(\boldsymbol{\xi}^T\boldsymbol{x})$. We need to show that $h \in \mathcal{F}$. But

$$
\begin{aligned}
h &= \nabla\phi^{-1}(\lambda\nabla\phi(f) + (1-\lambda)\nabla\phi(g))) \\
&= \nabla\phi^{-1}(\lambda\nabla\phi(\nabla\phi^{-1}(\boldsymbol{\theta}^T\boldsymbol{x})) + (1-\lambda)\nabla\phi(\nabla\phi^{-1}(\boldsymbol{\xi}^T\boldsymbol{x})))) \\
&= \nabla\phi^{-1}(\lambda\boldsymbol{\theta}^T\boldsymbol{x} + (1-\lambda)\boldsymbol{\xi}^T\boldsymbol{x}) \\
&= \nabla\phi^{-1}((\lambda\boldsymbol{\theta}^T + (1-\lambda)\boldsymbol{\xi}^T)\boldsymbol{x})
\end{aligned}
$$

which is again a GLM in $\mathcal{F}$. ∎

## B  Experimental details

We summarise our methodology to generate the illustrative experiments shown in the paper.

Full code is available at https://github.com/profgavinbrown/ondecompositions

We use a synthetic 1-d problem: $x \in [0, 15]$, and the true label is $y = x + 5\sin(2x) + \epsilon$, where $\epsilon$ is Gaussian noise with zero mean and $\sigma = 3$. Training data is $n = 100$ points, illustrated below.

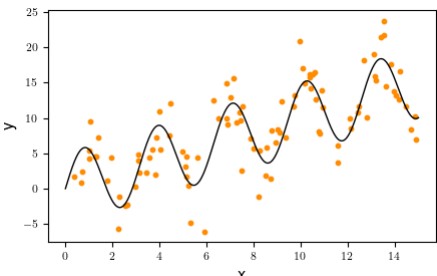

Figure 8: Synthetic problem for experiments.

Since this is a regression problem, $\ell(y, f(\boldsymbol{x})) = (y - f(\boldsymbol{x}))^2$, and $\mathring{f}_\phi(\boldsymbol{x}) := \mathbb{E}_D[\hat{f}(\boldsymbol{x})]$.

The function class $\mathcal{F}$ is defined as the set of all trained models obtained over $T = 1000$ independently sampled datasets, each of size $n = 100$. The best-in-class model is the minimum across the $T$ trials:

$$f^* := \arg\min_D \hat{R}(\hat{f}_D) \tag{61}$$

where the risk $R(f)$ is approximated by sample of uniformly sampled points at a resolution of 0.001, giving a total of $n = 15,000$ test points. To simplify analysis, we assume $\hat{f} = \hat{f}_{erm}$.

## C   Example Literature Conflating the Decompositions

We provide evidence from several sources that the decompositions are often conflated. This includes recent papers in NeurIPS, ICML, the Journal of Physics, Transactions on Neural Networks, and MIT lecture notes.

### C.1   Published work

In a popular online textbook, Daumé (2017, Section 5.9) states

> *"The trade-off between estimation error and approximation error is often called the bias/variance trade-off, where "approximation error" is "bias" and "estimation error" is "variance."*

In NeurIPS, Lee et al. (2022) state:

> *"Model selection is a fundamental task in supervised learning and statistical learning theory. Given a sequence of model classes, the goal is to optimally balance the approximation error (bias) and estimation error (variance)"*

In the Journal of the Royal Statistical Society, Mukherjee et al. (2023) state:

> *"A bound of this type"* [...] *"describes the 'bias-variance' or the 'approximation-estimation' trade-off"* [...] *"the 'bias' ('approximation') in (15) is zero and the 'variance' ('estimation') term determines the estimation error"*

In the Journal of Physics, Chen et al. (2022) state:

> *"To achieve low prediction error in supervised learning, the approximation-estimation trade-off (also known as the bias-variance trade-off) should be considered"*

In the Bulletin of the AMS, Cucker & Smale (2002) (1984 Google Scholar citations, Feb 2024) state

> *"Then, typically, the approximation error will decrease when enlarging H, but the sample error will increase. This latter feature is sometimes called the bias-variance trade-off"* [...] *"The 'bias' is the approximation error and the 'variance' is the sample error."*

In Machine Learning Journal, Barron (1994) (1050 Google Scholar citations, Feb 2024) states

> *" one can deal effectively with the total risk of the estimation of functions, including both the approximation error (bias) and the estimation error (variance)"*

In IEEE Trans. Neural Networks, Kwok & Yeung (1996) states

> *"This is however not the case for R in the absence of a bias term,"* [...] *"and thus the approximation error (i.e., bias) cannot be made as small as desired by trading variance.*

In IEEE Trans. Neural Networks, Lei et al. (2014) state

> *"To see this, we identify two factors determining the model's generalization performance by recalling the following bias-variance decomposition"* [...] *"The first term is often called the estimation error, while the second is the approximation error [24], [28]."*

In ISI *Stat*, Wang et al. (2021) state:

> *"Hence, we follow the conventional approximation–estimation decomposition (or bias–variance trade-off) to decompose the empirical norm"*

The next equation they state is the approximation-estimation decomposition for squared loss.

In the Journal of Ecological Modelling, Masson et al. (1999) state:

> *"The more flexible the model is, the greater is its ability to approach any function, but the more instable is the estimation problem from a finite amount of data. This is known as the approximation/estimation or bias/variance tradeoff."*

In a PhD thesis, Merckling (2021) states

> *"The two terms $E_{app}$ and $E_{est}$ constitute the approximation-estimation tradeoff (a.k.a. bias-variance tradeoff) where high bias is similar to high approximation error known as underfitting, and high variance is similar to high estimation error known as overfitting."*

Poggio et al. (2003) states:

> *"The decomposition of equation (12) is indirectly related to the well-known bias and variance decomposition in statistics."*

> *"More generally, however, there is a trade-off between minimizing the sample error and minimizing the approximation error—what we referred to as the bias-variance problem."*

Though technically correct, this is a slightly misleading use of language.

Niyogi & Girosi (1996), also in a 1995 MIT PhD thesis (Niyogi, 1995) state

> *"As the number of parameters (proportional to n) increases, the bias (which can be thought of as analogous to the approximation error) of the estimator decreases and its variance (which can be thought of as analogous to the estimation error) increases for a fixed size of the data set. Finding the right bias-variance trade-off is very similar in spirit to finding the trade-off between network complexity and data complexity."*

Wang & Lin (2023) states:

> *"These empirical findings deeply challenge the conventional wisdom that optimal generalization should be achieved by trading off bias (or approximation error) and variance (or estimation error)."*

In ICML 2023, Dubois et al. (2023)

> *"In supervised learning, one can get more fine-grained insights using the estimation/approximation (or bias/variance) risk decomposition,"*

> *"The estimation/approximation or the bias/variance decomposition has been very useful for practitioners and theoreticians to focus on specific risk components"*

though in the Appendix of the same article they state *"the approximation-estimation tradeoff (or the related bias-vias tradeoff)".* [**sic, including typo**]

Fan et al. (2021) state:

> *"We follow the conventional approximation-estimation decomposition (sometimes, also bias-variance tradeoff)*

The next equation they state is the approximation-estimation decomposition.

In a PhD thesis, Haury (2012) states:

> *Figure 1.3: Approximation error and estimation error. The error made when choosing [a model] can be seen as the sum of bias and variance. The bias refers to the approximation error and the variance to the estimation error.*

### C.2 Online materials conflating the decompositions

At the time of writing this article, all material was available at the URLs below. As these are not archived in perpetuity, we cannot guarantee availability in the future.

New York University:

> Slide 30 states Approximation error = "bias", and Estimation error = "variance".
> https://davidrosenberg.github.io/mlcourse/Archive/2016/Lectures/1b.
> intro-slt-riskdecomp.pdf

MIT:

> Module 9.520 lecture slides 17-19 use the title "Bias-Variance Tradeoff" but proceed to discuss the approximation-estimation decomposition.
> https://www.mit.edu/~9.520/fall18/slides/Class14_SL.pdf

University of Wisconsin:

> *"This decomposition into stochastic and approximation errors is similar to the bias-variance tradeoff which arises in classical estimation theory: the approximation error is like a bias squared term, and the estimation error is like a variance term."*
> https://nowak.ece.wisc.edu/SLT09/lecture3.pdf

Reddit Data Science discussion forum:

> *"A couple people have been confused by the exact terminology. I should clarify that bias-variance decomposition is technically different than the approximation-estimation error decomposition. But they are extremely similar, and* **in most cases they are mathematically equivalent. In fact, it is useful to think of the approximation-estimation decomp as a sub-case of the bias-variance decomposition***, if we make the assumption that our training algorithm is expected to output the best model in its hypothesis class. If this assumption can be made, then they become mathematically equivalent most intents and purposes. It's important to note that most modern class machine learning algorithms and classes satisfy this assumption, so they are equivalent."*
> https://www.reddit.com/r/datascience/s/ujFSz3rYqO

