# OpenReview forum: "Bias/Variance is not the same as Approximation/Estimation"
_TMLR — Accepted by TMLR_

### Review · Reviewer_hwRR · 2023-11-01

**Summary Of Contributions:**

The paper explores the connections and differences between the bias-variance and approximation-estimation decompositions, two fundamental and closely related decompositions in classical statistics and machine learning. In particular, the paper formalizes a generalized bias-variance decomposition under the Bregman divergence loss, and points out that the bias decomposes into approximation error and estimation bias, and hence is not the same as approximation error.

**Audience:**

Yes

**Claims And Evidence:**

Yes

**Requested Changes:**

The paper should not oversell the novelty and priority of the main results. Rather, the discussion should be focused on the new insights and their implications for machine learning theory.

**Strengths And Weaknesses:**

Strengths:

The paper does a good job in clarifying misconceptions regarding the two decompositions and provides some useful discussion on their misuse in machine learning.

Weaknesses:

The fact that the bias decomposes into approximation error and estimation bias, though less spelled out in the literature, is in fact well known in the statistical machine learning community, and was taught here in our university in introductory statistical learning courses. As the paper mentioned, the widely used textbook ESL discusses this point very clearly under the squared error loss. Some machine learning papers use bias and variance as synonyms for approximation error and estimation error, which differ from their classical meanings in statistics but are usually clear from the context. Similarly, the bias and variance defined in (11) (also, (14) and (15)) cannot be interpreted in the classical sense unless the squared error loss is adopted. Hence, I do not agree with the paper that these connections “have not been previously observed” or “not been explored before.” The proofs of the main results are elementary and straightforward.

---

> ### Author Response · Authors · 2023-11-01
> **Response to reviewer hwRR**
>
> Many thanks for the review.  A revision has been uploaded. We hope this addresses your concerns.
>
> > *The fact that the bias decomposes into approximation error and estimation bias, though less spelled out in the literature, is in fact well known in the statistical machine learning community.*
>
> We agree the paper has to tread a fine line - this is something that could be perceived as well-known, but we believe is only "obvious" once stated explicitly, and raises several non-intuitive issues.  We believe most of the results are not “well-known”, i.e. by the majority of the community, and would be of interest to them.
>
> > *and was taught here in our university in introductory statistical learning courses.*
>
> Can the reviewer post a specific (public) source of such information, dated before the public release of our paper?
>
> > *As the paper mentioned, the widely used textbook ESL discusses this point very clearly under the squared error loss.*
>
> This point, and how our work relates, was discussed extensively in appendix B.   To summarise here - the ESL book discusses “model bias”. At no point does it make the connection to approximation error.  Even if we believe this is implicit, it also makes no mention of estimation error, and the associated literature around uniform deviation bounds.  More critically, the book assumes such a decomposition *only* holds for a linear model - which we proved is not the case.  See appendix B for details.
>
> > *Some machine learning papers use bias and variance as synonyms for approximation error and estimation error, which differ from their classical meanings in statistics but are usually clear from the context.*
>
> This is the whole point of the paper - to remove such ambiguity, and show that this synonym/analogy has severe problems, which we make formal.  This is, we think, interesting to the ML community.
>
> > *Similarly, the bias and variance defined in (11) (also, (14) and (15)) cannot be interpreted in the classical sense unless the squared error loss is adopted.*
>
> We are not sure what the reviewer means by this point. Equations (11,14,15) all apply for a wide range of losses, beyond just squared loss.
>
> > *Hence, I do not agree with the paper that these connections “have not been previously observed” or “not been explored before.” The proofs of the main results are elementary and straightforward.*
>
> We don't believe the difficulty of a proof is an acceptance criterion for TMLR? We worked hard to simplify the paper’s proofs and overall logic.  It is certainly possible to have a more complex presentation, which we avoided.
>
> We are happy to tone down the claims to some degree. Does the reviewer have specific wording they would be ok with?

---

### Review · Reviewer_hLEj · 2023-11-04

**Summary Of Contributions:**

This paper considers the connection between two related concepts in statistics and machine learning, the bias-variance trade-off and the approximation-estimation decomposition. Both characterize the performance of a model as a function of the model size and thus are sometimes conflated. The authors rigorously characterize the connection between the two for the class of Bregman divergence losses, and make several observations: 1) the bias cannot clearly characterize model capacity, 2) estimation bias can be negative, and 3) the double descent phenomenon seen for deep networks is mostly a result of the estimation variance.

**Audience:**

Yes

**Broader Impact Concerns:**

This work is largely theoretical, so there are no broader impact concerns.

**Claims And Evidence:**

Yes

**Requested Changes:**

Would strengthen the work:
- It would make the paper more aesthetically pleasing to create the plots using Matplotlib (https://matplotlib.org/stable/)

**Strengths And Weaknesses:**

# Strengths #

I think that this work would be widely relevant as it connects concepts in two communities. The paper is written clearly and is straightforward to understand. After checking the math, the theoretical results are sound. The authors also provide examples for illustrations.

# Weaknesses #

It is not currently clear how the results in the paper would be applied in future work. The authors mention the importance of the centroid of the distribution in the conclusion, but do not expand on it. For example, how could the results be helpful for studying generalization theory?

---

> ### Author Response · Authors · 2023-11-06
> **Reply to Reviewer hLEj**
>
> Many thanks for the review and positive words. We have uploaded a revision.
>
> > *It is not currently clear how the results in the paper would be applied in future work. The authors mention the importance of the centroid of the distribution in the conclusion, but do not expand on it. For example, how could the results be helpful for studying generalization theory?*
>
> We have some ideas on this.  One might be the relation to algorithmic stability - we believe the  estimation variance is related to expressions we observe in that literature. Another might be expanding this to other loss families not currently covered, e.g. margin losses.
>
> In the revision we have discussed the literature for general losses where there is no bias-variance decomposition in the form commonly known (such as 0/1 loss) and how our work relates.  This again raises further issues that we have noted are worthy of further study.
>
> > *It would make the paper more aesthetically pleasing to create the plots using Matplotlib (https://matplotlib.org/stable/)*
>
> Figures 4/5 are Matplotlib. The other figures are in fact illustrations, drawn with another tool. Is this a "deal-breaker" for the reviewer? A requirement for acceptance?

---

### Review · Reviewer_F9ni · 2023-12-12

**Summary Of Contributions:**

This paper exams the connect and difference between the two types of
decomposition: the Bias/Variance decomposition and the Approximation/Estimation
decomposition. The two types of decomposition are defined on different
quantities related to the loss of estimating/predicting the responses. Some
theoretical results are provided to distinguish the two closely related but
different terms.

**Audience:**

Yes

**Claims And Evidence:**

Yes

**Requested Changes:**

Since the current paper is about the estimation/prediction of the response, it
is better to update the title to make it more accurate or extend the
investigation to include other types of statistical inferences. Inference on the
response is just one type of statistical inferences.

The bias/variance decomposition and the Approximation/Estimation are on
different quantities. It is better to state this more explicitly.

Poisson regression is a model not a divergence. It is better to call it Poisson
loss, divergence, or something else.

Theorem 1 is indeed trivial and may not be stated as a Theorem.

**Strengths And Weaknesses:**

Strengths:

The topic of the paper is interesting. Both decomposition are widely used across
different fields, but their connection and difference are not well documented.

The authors considered a general class of losses instead of focusing on
specifics.

Weaknesses:

The mathematical definitions and notations need improvements. For example, by
"the expectation $\mathbb{E}_{D}$ is over all possible training sets $D$", do
you mean the expectation is taken over the distribution of the training data
$D$? What is a "generator function"? "D" is first used to denote a sample but
used as a random variable in Definition 1.

The paper is about the estimation/prediction of the response, at least for the
bias/variance decomposition. The bias/variance decomposition does not apply in
general to parameter estimation, especially parameters in nonlinear models.

---

> ### Author Response · Authors · 2023-12-14
> **Reply to reviewer F9ni**
>
> Many thanks for the review and very detailed thoughts.   A revision has been uploaded addressing these points.
>
> > *The mathematical definitions and notations need improvements. For example, by "the expectation is over all possible training sets", do you mean the expectation is taken over the distribution of the training data? What is a "generator function"? "D" is first used to denote a sample but used as a random variable in Definition 1.*
>
> We have tightened-up the mathematics throughout.  Notation D is now a random variable everywhere - with distribution P(x,y)^n, i.e. samples of size n drawn i.i.d. from the data distribution. A generator function is terminology from the literature on Bregman divergences - we have added a footnote with a pointer to appropriate literature.
>
> > *The paper is about the estimation/prediction of the response, at least for the bias/variance decomposition. The bias/variance decomposition does not apply in general to parameter estimation, especially parameters in nonlinear models.*
>
> A sentence has been added to the introduction to clarify this: *"We note that this decomposition, as used in the Machine Learning literature, concerns inference of the response/target variable—and not of the parameters, as is more common in classical statistics."*
>
> > *The bias/variance decomposition and the Approximation/Estimation are on different quantities. It is better to state this more explicitly.*
>
> A sentence has been added at the start of Section 3: *``Perhaps the most obvious difference is that they are on different quantities—the excess risk of an ERM, versus the expected risk of an arbitrary trained model.''*
>
> > *Poisson regression is a model not a divergence. It is better to call it Poisson loss, divergence, or something else.*
>
> Agreed. Apologies. Changed to "Poisson loss".
>
> > *Theorem 1 is indeed trivial and may not be stated as a Theorem.*
>
> This has been changed to Proposition 1.

---

> > ### Comment · Reviewer_F9ni · 2023-12-27
> > **missing references**
> >
> > Thank you for your responses.
> >
> > After reading other reviewers' comments and the authors' responses, I realized
> > that there is a significant missing componenet of the current version.
> >
> > The paper lacks specific references to support the key assertion.  It is stated
> > in the Abstract that "It is commonly stated that they are ``closely related``,
> > or ``similar in spirit``", and in the Conclusions that "On a literature review,
> > we found numerous sources incorrectly stating the two were equivalent, or
> > related as a special case / general case.".  However, the paper does not provide
> > specific references to justify these statements.  As these claims underpin the
> > paper's motivation, a section for an extensive list of literature with the
> > misconceptions of equivalence or relation between the two subjects is essential.

---

> ### Author Response · Authors · 2023-12-28
> **missing references**
>
> We have uploaded a revision with a new appendix, giving several such examples.
>
> This includes peer-reviewed work: recent papers in NeurIPS and ICML, as well as papers from the 1990s with over 1000 citations.  This also includes widely referenced University course materials: from MIT, New York University, and Wisconsin.

---

### Author Response · Authors · 2024-02-13
**Reply to AE**

Many thanks for these final points.  A full revision has been uploaded.
We have highlighted (in red) all significant changes responding to these points.

> *Please remove claims that this is the first work to analyze the distinction between bias/variance and approximation error/estimation error.*

Agreed. The text has been modified accordingly.

> *the paper should be clearer that the main results are focused on Bregman divergences. This should be clear from the abstract.*

The main results are indeed focused on Bregman divergences.  However, we also considered the case of more general losses, in subsec 4.5.  This is now reflected in the abstract/introduction.

> *Please edit the abstract to use more professional language.*

Agreed. Modified accordingly.

> *right at the beginning of 2.1: nonstandard notation f: x -> y.*

Amended to f: \mathcal{X} -> \mathcal{Y}.

> *what is F_all? Not well defined.*

F_all is the space of measurable functions from \mathcal{X} to \mathcal{Y}. The text at the start of section 2.1 has been modified accordingly.

> *equation 4, 5, 7, etc not well defined; what is arg inf?.... [...] you could resolve issues re: arginf by assuming there exists a unique minimizer, but you'd need to find and cite conditions for when this is guaranteed from the literature.*

The arginf’s are necessary as it makes clear the minimizers are often non-unique, and a random variable. This is critical to the argument --- i.e., the estimation error is a random variable precisely because the f_{erm} is non-unique, and a random variable dependent on the data. We have tided up the language/notation, making this clear in the text.

> *after eq 5: what is a "sample drawn from a random variable"? I suspect you just mean that S is a realization of n iid draws from P(x,y)*

Indeed, this is exactly what we meant, as the notation P(x,y)^n was meant to indicate. The text around these equations has been modified.

> *there are at least a few things wrong with eq 11; ell is defined as mapping Y x ri(Y) to R+ [...]*

The divergence is defined over Y x ri(Y) to enable simple inclusion of the KL divergence between multinoullis, since ln(y) is non-differentiable at 0. We define it this way and assume differentiability on the interior, as is standard for Bregmans. This is now clearer via defining a Bregman divergence, now Definition 1. The argmin for the left centroid is over Y to match up with Nielsen & Nock, however, we are guaranteed that the centroid is in the interior - justification now in text.  For y^*, we removed the specific arginf to avoid confusion as the Bayes model is the same for any Bregman divergence -  E[Y|X] - see Banerjee2005a.

> *Definition 1 is not well-formed. For example, the argmin is not always guaranteed to exist, and even if it does it may not be unique. There needs to be more conditions on ell, f, and D.*

We have removed this, and relegated the concept to the discussion. There we assume it exists, but it may not be unique. We have given examples where this is the case.

> *For Defn 2, please cite a specific result from Nielsen and Nock that guarantees that it is well-formed.*

Cited Theorem 3.2 in Nielsen & Nock (2008). See also Nielsen & Nock (2020), as a correction to the 2008 paper.

> *there are a few places (e.g., with Amari as well) where long manuscripts / books are cited for a single result. Please provide specific citations of individual results, pages, etc everywhere*

We have addressed this throughout, e.g., Amari (2008, Eq 32) for a dual-convex set.  If there any further points like this, we are happy to cite a specific result.

> *Please confirm that Eq 13 is strictly true, i.e. the LHS and RHS are always not equal. Otherwise that inequality is not correct.*

The bias-variance decomposition does not hold for the 0/1 loss in general, as has been confirmed with toy counterexamples by multiple authors, which we did cite.  However, we agree there may be pathological edge cases. To avoid an incorrect claim we have removed the inequality, and referred simply to the mass of literature proposing alternative 0/1 loss decompositions.

> *Proposition 1 shouldn't be a proposition; it's just adding and subtracting R(fcirc). No need to have a stated result here.*

We have changed this to a definition. This is necessary for referencing within the document, and to highlight how the terms relate to one another (in the discussion section).  We obviously are claiming no degree of novelty or insight from this trivial observation, but it is critical to the narrative to have it highlighted.

> *I'm not sure why E_est(v) and E_est(b) are defined after Prop 1 [...] Similarly, E_app is not defined.*

The symbols were used in section 4, and in the appendix. We have made this explicit at the top of section 4, and removed their use around Prop 1 (now Def 1).

---

> ### Comment · Action_Editor_mSEG · 2024-02-19
> **Please upload your camera ready asap**
>
> On my side it looks like your camera ready has not been uploaded. If you believe it has been uploaded already, please let me know and I'll get in touch with TMLR website staff.

---

> > ### Author Response · Authors · 2024-02-19
> > **Done**
> >
> > Apologies - did not see the request for camera ready.   Now uploaded.

---

### Decision · Action_Editor_mSEG · 2024-01-20

**Recommendation:** Accept with minor revision

**Comment:**

Given the assessments of reviewers, I recommend acceptance with minor revisions; details follow.

Please remove claims that this is the first work to analyze the distinction between bias/variance and approximation error/estimation error. There isn't really a reason to make such claims, and it doesn't detract from the work to remove them. Furthermore, there is not compelling evidence presented to support the claim (it would indeed be very difficult to collect such evidence, and I'm not sure I believe the claim myself without making it far more specific).

Along those lines, the paper should be clearer that the main results are focused on Bregman divergences. This should be clear from the abstract. The paper does indeed provide a nice general view of bias/variance/approximation/estimation, but it is not arbitrarily general.

Please edit the abstract (and the rest of the paper) to use more professional language. Things like "spoiler: no they're not" aren't appropriate for formal academic publications.

There are numerous minor mathematical mistakes and imprecise/unclear statements that need to be fixed before this is published. None of these are major errors in the reasoning of the paper, but if the paper claims to be providing a more careful analysis of imprecise claims in the community, it had better be quite precise itself.

- right at the beginning of 2.1: nonstandard notation f: x -> y. One would think f is a function from R^d to R^k given your setting, so should be f:R^d -> R^k
- what is F_all? Not well defined.
- equation 4, 5, 7, etc not well defined; what is arg inf?
    - a general comment here after doing a pass over the whole paper: the paper is generally very imprecise about arg minimizers. Previous work generally avoids these technicalities entirely by just referring to inf R(...), as opposed to attempting to handle the minimizers explicitly, but I suspect that isn't possible everywhere in this paper.
- you could resolve issues re: arginf by assuming there exists a unique minimizer, but you'd need to find and cite conditions for when this is guaranteed from the literature.
- why use y* for a classifier in eq 4? Why not just completely avoid the abuse of notation and use f*_all or something of that sort?
- after eq 5: what is a "sample drawn from a random variable"? I suspect you just mean that S is a realization of n iid draws from P(x,y) but might be wrong. You do not need two separate symbols (S, D) for this.
- there are at least a few things wrong with eq 11; ell is defined as mapping Y x ri(Y) to R+, but the bias term appears to take an element of ri(Y) x Y. Defining y* to be the argmin over a relative interior does not make sense unless the inf is guaranteed to be in the relative interior, and then you can just argmin over the whole set anyway because the boundary ceases to be an issue.
- it is odd that the centroid is used in Eq 11 before it is defined later in Defn1 & 2
- Definition 1 is not well-formed. For example, the argmin is not always guaranteed to exist, and even if it does it may not be unique. There needs to be more conditions on ell, f, and D.
- For Defn 2, please cite a specific result from Nielsen and Nock that guarantees that it is well-formed.
    - there are a few places throughout (e.g., with Amari as well) where long manuscripts / books are cited for a single result. Please provide specific citations of individual results, pages, etc everywhere possible.
- Please confirm that Eq 13 is strictly true, i.e. the LHS and RHS are *always* not equal. Otherwise that inequality is not correct.
- Proposition 1 shouldn't be a proposition; it's just adding and subtracting R(fcirc). No need to have a stated result here.
- I'm not sure why E_est(v) and E_est(b) are defined after Prop 1. They don't appear to be used anywhere. If they only appear in the appendix, define them there near their first usage.
- Similarly, E_app is not defined.

**Audience:**

Yes; the subset of the TMLR audience working on predictive methods (classification, regression, etc) may find this work interesting. Most work in this area tends to focus on the squared error loss; this work generalizes the story somewhat to Bregman divergences. The results themselves are straightforward, but this is not a criterion for acceptance at TMLR.

**Claims And Evidence:**

The paper makes the following claims:
- there are a number of published works / trusted sources that conflate the decomposition of risk as "bias/variance" and "approximation error/estimation error"
- this paper presents a study of the relationship between these two decompositions
- this is the first work to discuss the connection of the two decompositions
- various theoretical results along those lines